

# Associations of PM$_{2.5}$ and its components with term preterm rupture of membranes: a retrospective study

Jiangxia Qin[1,*], Weiling Liu[2,*], Haidong Zou[1], Chong Zeng[3], Cifeng Gao[4] and Weiqi Liu[5]

[1] Department of Obstetrics, The Maternal and Children Health Care Hospital (Huzhong Hospital) of Huadu, Guangzhou, China

[2] Department of Clinical Laboratory, Foshan Fosun Chancheng Hospital, Guangzhou, China

[3] Department of Microbiology and Immunology; Institute of Geriatric Immunology; School of Medicine, Jinan University, Guangzhou, China

[4] Personnel Section, The Maternal and Children Health Care Hospital (Huzhong Hospital) of Huadu, Guangzhou, China

[5] Department of Clinical Laboratory, The Maternal and Children Health Care Hospital (Huzhong Hospital) of Huadu, Guangzhou, China

[*] These authors contributed equally to this work.

Corresponding authors
Cifeng Gao, 343102206@qq.com
Weiqi Liu, lwq_8103@163.com

## ABSTRACT

**Background.** There is evidence that fine particulate matter (PM$_{2.5}$) exposure is associated with premature rupture of membranes (PROM); however, studies of its effect on term PROM (TPROM) are limited, and the results are inconsistent.

**Objective.** This study aimed to investigate the association between exposure to PM$_{2.5}$ and its components and the risk of TPROM.

**Methods.** From 2018 to 2022, we collected delivery data from pregnant women in Guangzhou. Using 1:1 case matching, we included 1,216 TPROM cases and 1,216 controls. PM$_{2.5}$ and its component concentrations were obtained from Tracking Air Pollution in China. The time-varying mean concentration method was used to estimate exposure to PM$_{2.5}$ and its components during different trimesters. Cox proportional hazards models were used to calculate hazard ratios (HRs) and 95% confidence intervals (CIs) to evaluate the associations of exposure to PM$_{2.5}$ and its components with the risk of TPROM.

**Results.** The incidence of TPROM in this study was 19.66%. After adjusting for potential confounders, statistically significant associations were found between TPROM and exposure to PM$_{2.5}$, nitrate (NO$_3^-$), ammonium (NH$_4^+$), and black carbon (BC) during the second trimester and between TPROM and exposure to PM$_{2.5}$, sulphate (SO$_4^{2-}$), and BC during the third trimester. Specifically, the interquartile range (IQR) 3 (IQR3) and IQR4 of SO$_4^{2-}$ exposure during the third trimester increased the risk of TPROM by 18% (95% CIs [1.01–1.39]) and 18% (95% CIs [1.01–1.39]), respectively. A nonlinear relationship was observed between exposure to PM$_{2.5}$, SO$_4^{2-}$, NH$_4^+$, and OM during the second trimester and the risk of TPROM. No significant interactions were found between PM$_{2.5}$ and its components with TPROM across various subgroups.

**Conclusion.** Our findings indicate significant associations between the risk of TPROM and exposure to PM$_{2.5}$ and several of its components during pregnancy. Contribute to the literature on the associations of PM$_{2.5}$ and its components with TPROM.

# INTRODUCTION

Premature rupture of membranes (PROM) refers to the breaking of foetal membranes before labour begins, with no sign of labour present after one hour (*Committee on Practice Bulletins-Obstetrics, 2018*). PROM can be divided into preterm and term PROM (TPROMs). Preterm PROM typically occurs before 37 weeks of gestation, and TPROM are defined as the rupture of membranes that occurs from 37 weeks of gestation to the start of labour (*Endale et al., 2016*). Approximately 8% of pregnant women at term experience PROM (*American College of Obstetricians and Gynecologists' Committee on Practice Bulletins—Obstetrics, 2016*). The incidence of PROM in the United States is 5% (*Getahun et al., 2007*); in Nepal, it is 8% (*Prasad Dwa, Bhandari & Bajracharya, 2023*); and in China, the incidence of PROM is 18.72% (*Zhuang et al., 2020*). Despite the widespread concern about preterm PROMs, TPROMs are also a pregnancy complication that should not be ignored, and their occurrence can lead to adverse outcomes such as perinatal death, increased rates of caesarean section, early-onset neonatal pneumonia, and neonatal sepsis (*Herbst & Kallen, 2007*; *Middleton et al., 2017*; *Namli Kalem et al., 2017*; *Zhuang et al., 2022*).

Although the causes of PROM remain unclear, several factors have been implicated, including age, previous preterm birth, smoking, polyhydramnios, urinary and sexually transmitted infections, previous PROM, caesarean section and cervical incompetence (*ACOG Committee on Practice Bulletins-Obstetrics, 2007*; *Kaye, 2001*; *Kilpatrick et al., 2006*; *Puji Astuti, Ariyani & Mahayati, 2022*). Exposure to fine particulate matter (PM$_{2.5}$) increases the risk of PROM, according to the findings of recent investigations into the relationship between air pollutants and PROM (*Dadvand et al., 2014*; *Han et al., 2020*; *Ren et al., 2024*; *Zhang et al., 2021*). A retrospective study conducted in Anhui, China, involving 4,276 participants reported that for each 10 μg/m$^3$ increase in PM$_{2.5}$ exposure, the risk of PROM increased by 48% (95% confidence interval (CI) [1.16–1.89]) (*Yang et al., 2024*). A retrospective cohort study conducted in southern California revealed that mothers exposed to higher levels of PM$_{2.5}$ during pregnancy had an increased risk of PROM associated with heatwaves (*Jiao et al., 2023b*). A meta-analysis by *Liang et al. (2024)* indicated that PM$_{2.5}$ exposure during mid-pregnancy and short-term maternal exposure to PM$_{2.5}$ are associated with an increased risk of PROM. Therefore, PM$_{2.5}$ exposure during pregnancy is closely associated with PROM.

The aim of this study was to analyse data from pregnant women delivering at the Maternal and Children Health Care Hospital of Huadu in Guangzhou from 2018–2022, with a focus on the associations between exposure to PM$_{2.5}$ and its components and TPROM during pregnancy.
## METHODS

### Subjects

This retrospective study involved pregnant women who delivered at the Maternal and Children Health Care Hospital of Huadu in Guangzhou from November 2018 to December 2022. Data were collected using the hospital's electronic medical record management system. Relevant information, including age, occupation, ethnicity, blood type, delivery date, and clinical diagnosis, was obtained by reviewing the electronic medical records. Diagnoses were classified according to the ICD-10 codes. The exclusion criteria for the study participants were as follows: (1) residing outside Guangzhou; (2) *in vitro* fertilisation; (3) pregnancy of twins; (4) diabetes before pregnancy; (5) hypertension before pregnancy; and (6) insufficient data (Fig. 1). The study focused on subjects diagnosed with PROM (ICD-10 codes O42.000, O42.000 × 001, O42.100, O42.100 × 011, O42.200, O42.900), designated the disease group, with the other subjects forming the control group. This study received approval from the Ethics Committee of the Maternal and Children's Health Care Hospital of Huadu (approval no. 2024-001). The committee waived the requirement for informed consent because anonymised data were used. This study complied with the ethical principles outlined in the 1975 Declaration of Helsinki.

### Assessment of air pollution concentrations

We obtained the spatially continuous, grid-based daily average concentrations of $PM_{2.5}$ and its constituents from 2018–2022 from tracking air pollution (TAP) in China (http://tapdata.org.cn/). The TAP is a near-real-time air pollutant database that integrates information from ground observations, satellite aerosol optical depth, operational chemical transport model simulations and meteorological fields. It is based on a two-stage machine learning model that uses synthetic minority oversampling techniques and tree-based gap-filling estimation methods and has provided $PM_{2.5}$ data at a 10 × 10 km resolution since 2000 (*Geng et al., 2021*). On the basis of the 10 × 10 km resolution dataset, improving the dust emission module in the Community Multiscale Air Quality Modeling System and using the XGBoost algorithm to adjust the relative contributions of the $PM_{2.5}$ component concentrations ultimately provides more accurate 10 × 10 km resolution data for sulphate ($SO_4^{2-}$), nitrate ($NO_3^-$), ammonium ($NH_4^+$), organic matter (OM), and black carbon (BC) (*Liu et al., 2022*).

We used the time-varying average concentration approach to assess the relationships between exposure to $PM_{2.5}$ and its components and different stages of pregnancy, which were based on the date of birth and the last menstrual period. Following previous research (*Gong et al., 2022*), we determined the mean exposure concentrations for three specific intervals: the first trimester (weeks 1–12 of pregnancy; T1), the second trimester (weeks 13–27; T2) and the third trimester (T3), which was defined as the period from the 28th week until the birth of the baby.

### Covariates

For this study, covariates were selected on the basis of prior research (*Jena et al., 2022*; *Muniz Rodriguez, Pastor & Fox, 2021*; *Wallace et al., 2016*) and data available from the

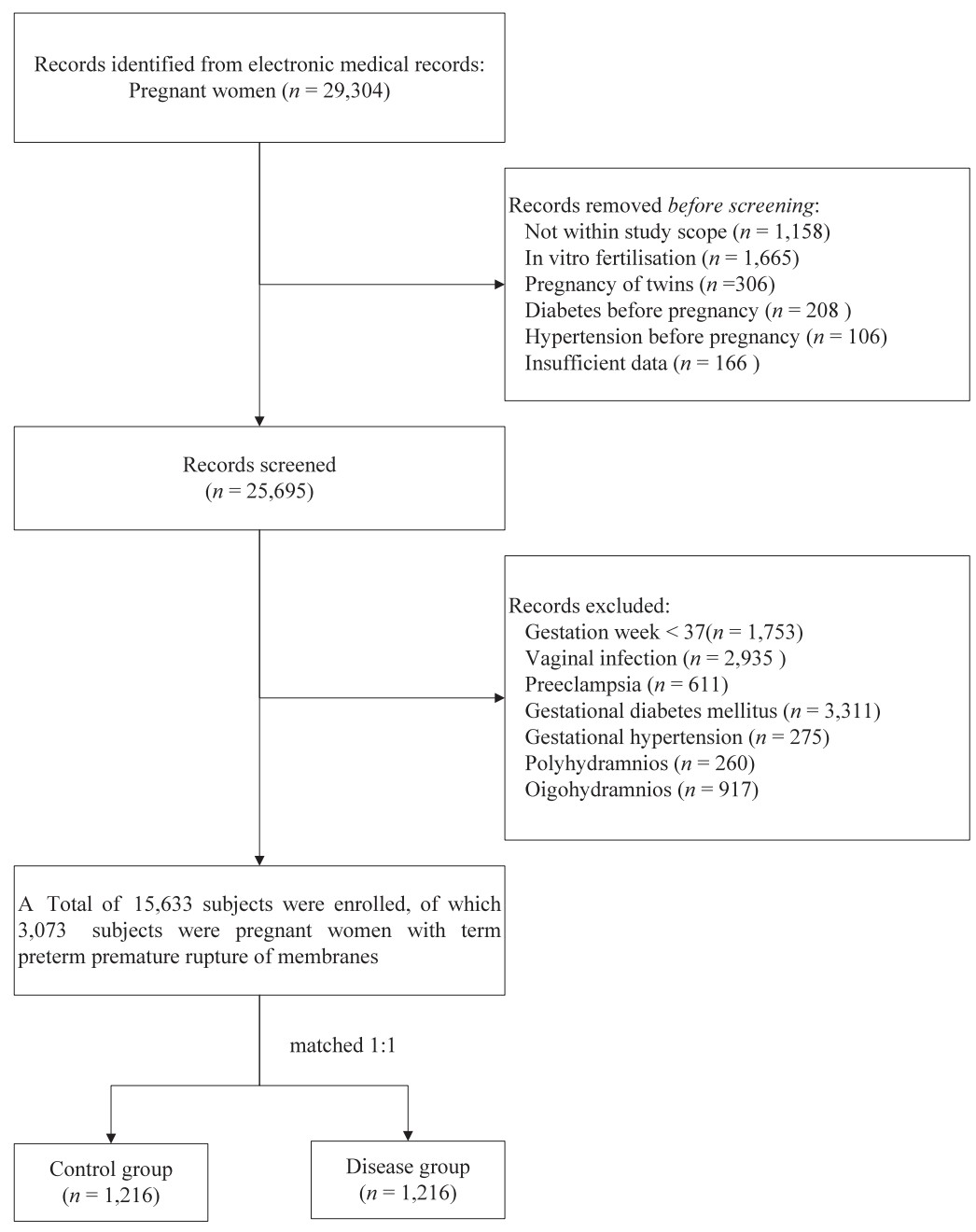

**Figure 1 Flowchart of participant screening.**

hospital's electronic medical record system. The covariates were age, ethnicity, occupation, blood type, primary, anaemia, adverse reproductive history, uterine scar, hepatitis B virus (HBV) status, and obesity during pregnancy.

The participants provided self-reported information regarding their ethnicity (such as Han, Hui, Miao, and Tujia), occupation category (such as employee, civil servant, professional, self-employed, farmer, and unemployed), marital status (married or

divorced), and blood type (A, B, O, and AB), and the study focused on women who were pregnant and gave birth for the first time. For further analysis, ethnicity was reclassified into Han or other, occupation type was regrouped as employed, self-employed, or other, and infant weight was stratified on the basis of recorded birth weight into low birth weight (<2,500 g), normal birth weight (2,500–4,000 g), or macrosomia (>4,000 g). Hepatitis B virus status indicates that a subject is a carrier of HBV. Obesity during pregnancy was defined as a body mass index greater than or equal to 28.0 kg/m² during pregnancy. An adverse reproductive history refers to a history of previous poor pregnancy outcomes or obstetric complications.

## Diagnosis of PROM

TPROM was determined based on the clinical diagnoses in the electronic medical records system. The diagnosis of PROM is based on three specific indicators observed by the clinician during a sterile speculum examination: (1) accumulation of clear fluid in the posterior vaginal fornix or exudation of fluid from the cervical orifice; (2) alkaline pH of the cervicovaginal secretions, usually detected by a change in colour from yellow to blue on nitrozine paper; and/or (3) the appearance of fernlike patterns in the dry cervicovaginal secretions under the microscope (*Caughey, Robinson & Norwitz, 2008*; *Wang et al., 2019*).

## Statistical analyses

The descriptive characteristics of the participants were analysed *via* nonparametric tests for continuous variables and the chi-square test ($\chi^2$) for categorical variables. This retrospective study used a case-control design, with cases being TPROM patients and controls being non-TPROM patients. Patients and controls were matched 1:1 for age and last menstrual period (LMP). This matching strategy aimed to ensure that all participants had the same starting point in early pregnancy, thereby placing each in a similar environment for most of their pregnancies. Once matching was complete, a Cox proportional hazards model was used to estimate hazard ratios (HRs) and 95% confidence intervals (CIs) to assess the associations of exposure to $PM_{2.5}$ and its components with the risk of TPROM, controlling for potential confounders, including age, occupation, blood group, ethnicity, parity, HBV status, obesity during pregnancy, adverse reproductive history, and uterine scar. Based on previous studies (*Desquilbet & Mariotti, 2010*; *Fan, Zhang & Zhong, 2017*; *Wan et al., 2024*), we used Cox proportional hazards model alongside restricted cubic splines curves to assess the relationship between exposure to $PM_{2.5}$ and its constituents and the risk of TPROM. The reference value, established at the 10th percentile with an HR of 1, was compared using knots positioned at the 5th, 35th, 65th, and 95th percentiles of the concentration levels of $PM_{2.5}$ and its components. In addition, interaction analyses were performed to explore possible interactions between exposure to air pollutants during pregnancy and these confounders. Matching was performed *via* R version 4.2.2, and regression analyses were performed using STATA 16.0. A *p* value of less than 0.05 was considered indicative of statistical significance.

## RESULTS

### Baseline characteristics

In a sample of 15,633 cases, 19.66% of full-term pregnant women experienced TPROM, using 1:1 case matching, we included 1,216 TPROM cases and 1,216 controls. After matching, the $Z$-values or $\chi^2$-values for age, ethnicity, occupation, primary, anaemia, adverse reproductive history, uterine scar, Hepatitis B (HBV) status, infant weight, and pollution concentration all decreased to varying degrees. Specifically, age, race, occupation and adverse obstetric history changed significantly before and after matching, with $p$-values of 0.025, 0.016, <0.001 and 0.017 before matching and 1,000, 0.169, 0.777 and 0.205 after matching, respectively (Table 1), indicating that the differences between the case and control groups were significantly reduced. This suggests that case matching helps balance baseline characteristics, enhancing the accuracy of the assessment of factors associated with TPROM.

### Correlations among $PM_{2.5}$, $SO_4^{2-}$, $NO_3^-$, $NH_4^+$, OM, and BC

Table S1 displays the spearman correlation coefficients for mean daily concentrations of $PM_{2.5}$ and its components. $PM_{2.5}$ showed strong positive correlations with all of its components, ranging from 0.933 (with $NO_3^-$) to 0.990 (with OM). $SO_4^{2-}$, $NO_3^-$, $NH_4^+$, OM, and BC also exhibited high correlations, indicating strong associations among these pollutants.

### Association of air pollution with PROM

After adjusting for potential confounders, we observed no statistically significant HR for the association of TPROM with exposure to $PM_{2.5}$ or its components during the first trimester (Table 2). During the second trimester, we observed a statistically significant association between TPROM and exposure to $PM_{2.5}$, $NO_3^-$, $NH_4^+$, and BC. Specifically, increases in the interquartile range (IQR) 2 of $PM_{2.5}$, $NO_3^-$, and $NH_4^+$ were associated with hazard ratios of 1.19 (95% CI [1.01–1.40]), 1.19 (95% CI [1.02–1.40]), and 1.21 (95% CI [1.03–1.42]), respectively. Additionally, a significant association was observed with an increase in IQR3 for BC, with an HR of 1.18 (95% CI [1.01–1.39]) (Table 3). In the third trimester, TPROM was significantly associated with exposure to $PM_{2.5}$, $SO_4^{2-}$, and BC. Specifically, the IQR3 and IQR4 of $SO_4^{2-}$ exposure during the third trimester increased the risk of TPROM by 18% (95% CIs [1.01–1.39]) and 18% (95% CIs [1.01–1.39]), respectively (Table 4).

After adjusting for covariates, a nonlinear relationship was observed between exposure to $PM_{2.5}$, $SO_4^{2-}$, $NH_4^+$, and OM during the second trimester and the risk of TPROM ($p$ values for nonlinearity of 0.019, 0.044, 0.035, and 0.034, respectively) (Fig. 2).

### Subgroup analysis

In subgroup analyses after adjustment for confounders, we observed no significant interactions between exposure to $PM_{2.5}$, $SO_4^{2-}$, $NO_3^-$, $NH_4^+$, OM, or BC and TPROM during pregnancy in terms of age, occupation, ethnicity, parity, HBV status, adverse reproductive history, or uterine scar subgroup (Table 5 and Fig. 3).

**Table 1 Descriptive characteristics of this study, 2018–2022.**

| Variable | Before matching | | | | After matching | | | |
|---|---|---|---|---|---|---|---|---|
| | Control group ($n = 12{,}560$) | Disease group ($n = 3{,}073$) | Z/$\chi^2$ | p-value | Control group ($n = 1{,}216$) | Disease group ($n = 1{,}216$) | Z/$\chi^2$ | p-value |
| Age, mean (SD) | 29.30 (4.59) | 29.2 (4.47) | 2.238 | 0.025 | 28.8 (3.26) | 28.8 (3.26) | 0.000 | 1.000 |
| Ethnicity[a], n (%) | □ | □ | 5.755 | 0.016 | □ | □ | 1.894 | 0.169 |
| Han | 12,305 (98.0%) | 2,989 (97.3%) | | | 1,194 (98.2%) | 1,184 (97.4%) | | |
| Other | 255 (2.03%) | 84 (2.73%) | | | 22 (1.81%) | 32 (2.63%) | | |
| Occupation[b], n (%) | □ | □ | 19.100 | <0.001 | □ | □ | 0.505 | 0.777 |
| Employed | 7,300 (58.1%) | 1,888 (61.4%) | | | 729 (60.0%) | 739 (60.8%) | | |
| Self-employed | 788 (6.27%) | 139 (4.52%) | | | 61 (5.02%) | 54 (4.44%) | | |
| Other | 4,472 (35.6%) | 1,046 (34.0%) | | | 426 (35.0%) | 423 (34.8%) | | |
| Blood Type, n (%) | □ | □ | 1.345 | 0.719 | □ | □ | 2.347 | 0.504 |
| Type A | 3,500 (27.9%) | 856 (27.9%) | | | 326 (26.8%) | 338 (27.8%) | | |
| Type B | 3,134 (25.0%) | 763 (24.8%) | | | 302 (24.8%) | 271 (22.3%) | | |
| Type O | 5,053 (40.2%) | 1,257 (40.9%) | | | 503 (41.4%) | 524 (43.1%) | | |
| Type AB | 873 (6.95%) | 197 (6.41%) | | | 85 (6.99%) | 83 (6.83%) | | |
| Primary, n (%) | □ | □ | 71.117 | <0.001 | □ | □ | 24.440 | <0.001 |
| NO | 4,692 (37.4%) | 898 (29.2%) | | | 401 (33.0%) | 291 (23.9%) | | |
| Yes | 7,868 (62.6%) | 2,175 (70.8%) | | | 815 (67.0%) | 925 (76.1%) | | |
| Anemia, n (%) | □ | □ | 0.450 | 0.502 | □ | □ | 0.279 | 0.597 |
| No | 5,958 (47.4%) | 1,437 (46.8%) | | | 562 (46.2%) | 575 (47.3%) | | |
| Yes | 6,602 (52.6%) | 1,636 (53.2%) | | | 654 (53.8%) | 641 (52.7%) | | |
| Adverse reproductive history, n (%) | □ | □ | 5.699 | 0.017 | □ | □ | 1.798 | 0.205 |
| No | 11,394 (90.7%) | 2,830 (92.1%) | | | 1,109 (91.2%) | 1,127 (92.7%) | | |
| Yes | 1,166 (9.28%) | 243 (7.91%) | | | 107 (8.80%) | 89 (7.32%) | | |

**Table 1** (*continued*)

| Variable | Before matching | | | | After matching | | | |
|---|---|---|---|---|---|---|---|---|
| | **Control group (*n* = 12,560)** | **Disease group (*n* = 3,073)** | **Z/χ²** | **p-value** | **Control group (*n* = 1,216)** | **Disease group (*n* = 1,216)** | **Z/χ²** | **p-value** |
| Uterine scar, *n* (%) | ☐ | ☐ | 254.427 | <0.001 | ☐ | ☐ | 148.632 | <0.001 |
| No | 10,003 (79.6%) | 2,825 (92.0%) | | | 915 (75.2%) | 1,134 (93.3%) | | |
| Yes | 2,557 (20.4%) | 247 (8.04%) | | | 301 (24.8%) | 82 (6.74%) | | |
| HBV status, *n* (%) | ☐ | ☐ | 7.354 | 0.007 | ☐ | ☐ | 4.433 | 0.042 |
| No | 11,511 (91.6%) | 2,862 (93.1%) | | | 1,106 (91.0%) | 1,134 (93.3%) | | |
| Yes | 1,049 (8.35%) | 211 (6.87%) | | | 110 (9.05%) | 82 (6.74%) | | |
| Obesity in pregnancy, *n* (%) | ☐ | ☐ | 0.157 | 0.692 | ☐ | ☐ | 2.586 | 0.108 |
| No | 12,486 (99.4%) | 3,053 (99.3%) | | | 1,206 (99.2%) | 1,212 (99.7%) | | |
| Yes | 74 (0.59%) | 20 (0.65%) | | | 10 (0.82%) | 4 (0.33%) | | |
| Infant gender, *n* (%) | ☐ | ☐ | 0.162 | 0.688 | ☐ | ☐ | 0.200 | 0.655 |
| Male | 6,685 (53.2%) | 1,648 (53.6%) | | | 649 (53.4%) | 660 (54.3%) | | |
| Female | 5,875 (46.8%) | 1,425 (46.4%) | | | 567 (46.6%) | 556 (45.7%) | | |
| Infant Weight, *n* (%) | ☐ | ☐ | 84.714 | <0.001 | ☐ | ☐ | 13.557 | 0.001 |
| <2,500 g | 351 (2.79%) | 176 (5.73%) | | | 38 (3.12%) | 57 (4.69%) | | |
| 2,500–4,000 g | 11,945 (95.1%) | 2,871 (93.4%) | | | 1,150 (94.6%) | 1,150 (94.6%) | | |
| >4,000 g | 264 (2.10%) | 26 (0.85%) | | | 28 (2.30%) | 9 (0.74%) | | |
| Pollution, median (IQR)[c] | ☐ | ☐ | | | ☐ | ☐ | | |
| $PM_{2.5}$ (μg/m³) | 27.8 (21.9, 29.8) | 27.7 (21.7, 29.7) | 1.173 | 0.241 | 28.10 (22.10, 29.80) | 28.10 (22.10, 29.80) | −0.060 | 0.950 |
| $SO_4^{2-}$ (μg/m³) | 5.4 (4.28, 5.88) | 5.39 (4.27, 5.87) | 1.103 | 0.270 | 5.46 (4.33, 5.88) | 5.46 (4.32, 5.88) | −0.060 | 0.952 |
| $NO_3^-$ (μg/m³) | 3.81 (3.20, 4.38) | 3.77 (3.12, 4.36) | 1.662 | 0.097 | 3.82 (3.22, 4.41) | 3.82 (3.20, 4.41) | −0.001 | 0.999 |

**Table 1** (*continued*)

| Variable | Before matching | | | | After matching | | | |
|---|---|---|---|---|---|---|---|---|
| | Control group ($n = 12,560$) | Disease group ($n = 3,073$) | $Z/\chi^2$ | *p*-value | Control group ($n = 1,216$) | Disease group ($n = 1,216$) | $Z/\chi^2$ | *p*-value |
| $NH_4^+$ ($\mu g/m^3$) | 2.91 (2.39, 3.28) | 2.9 (2.35, 3.26) | 1.619 | 0.106 | 2.92 (2.42, 3.28) | 2.93 (2.40, 3.29) | −0.048 | 0.962 |
| OM ($\mu g/m^3$) | 7.38 (5.72, 7.97) | 7.36 (5.71, 7.94) | 1.015 | 0.310 | 7.47 (5.75, 7.97) | 7.48 (5.74, 7.97) | −0.063 | 0.950 |
| BC ($\mu g/m^3$) | 1.52 (1.19, 1.64) | 1.52 (1.19, 1.64) | 0.896 | 0.370 | 1.54 (1.20, 1.64) | 1.54 (1.20, 1.65) | −0.075 | 0.940 |

**Notes.**
[a] Han, Hui, Miao, Tujia, *etc.*
[b] Employee, civil servant, professional, self-employed, farmer, unemployed, *etc.*
[c] Median (IQR) for exposure during first to third trimester.
HBV status, hepatitis B virus status; IQR, interquartile range.
$PM_{2.5}$, particulate matter with aerodynamic diameter of $\leq 2.5$ μm; $SO_4^{2-}$, sulfate; $NO_3^-$, nitrate.
$NH^+$, ammonium; OM, organic matter; BC, black carbon.

## DISCUSSION

In a retrospective study, we investigated the associations of exposure to $PM_{2.5}$ and its components during pregnancy with TPROM. We found that exposure to $PM_{2.5}$, $SO_4^{2-}$, $NO_3^-$, $NH_4^+$, and BC was significantly associated with an increased risk of TPROM. However, when we conducted subgroup analyses by age, ethnicity, occupation, blood type, parity, adverse reproductive history, and uterine scarring, we did not observe any significant interactions between exposure to $PM_{2.5}$ and its components and TPROM. The findings suggest that exposure to air pollution during pregnancy increases the risk of TPROM irrespective of individual characteristics.

An increasing number of studies indicate that exposure to respirable particulate matter, especially $PM_{2.5}$, is closely related to oxidative stress responses (*Ambroz et al., 2016*; *Moller & Loft, 2010*; *Orellano et al., 2020*; *Saffari et al., 2014*). Long-term exposure to $PM_{2.5}$ can inhibit nitric oxide (NO)-dependent microvascular dilation and impair mitochondrial oxidative capacity (*Della Guardia & Wang, 2023*). Because mitochondria are prone to accumulate oxidative DNA modifications in oxidative environments (*Muftuoglu, Mori & De Souza-Pinto, 2014*), this may affect the function and integrity of foetal membranes, potentially leading to PROM. A study conducted in Taiyuan, China, revealed a significant association between oxidative stress and exposure to $PM_{2.5}$ and its components (*Li et al., 2023*). *Kumari et al. (2023)* reported in a case-control study that the mtDNA copy number was significantly greater in patients with early membrane rupture than in controls. In addition, *Menon et al. (2012)* conducted a study in the United States on telomere length in foetal leukocytes associated with preterm-PROM compared with intact membranes in preterm and term births. They reported that telomere length was significantly shorter in preterm-PROM cases than in age-matched preterm controls. Therefore, exposure to $PM_{2.5}$ during pregnancy may contribute to PROM.

The incidence of TPROM in this study was 19.66%, which is consistent with the 19.55% incidence rate reported in Shanghai (*Li et al., 2021*). This similarity may be attributed

**Table 2** HRs and 95% CIs associated with TPROM in the first trimester per IQR increase in $PM_{2.5}$ and its components.

| Variable | | Crude | | Adjusted[a] | |
|---|---|---|---|---|---|
| | | HRs (95% CIs) | *p*-value | HRs (95% CIs) | *p*-value |
| $PM_{2.5}$ | IQR1 | Ref (−) | | Ref (−) | |
| | IQR2 | 0.89 (0.76–1.04) | 0.138 | 0.88 (0.74–1.03) | 0.105 |
| | IQR3 | 1.02 (0.87–1.20) | 0.785 | 1.03 (0.88–1.21) | 0.713 |
| | IQR4 | 1.12 (0.96–1.31) | 0.162 | 1.10 (0.94–1.29) | 0.249 |
| $SO_4^{2-}$ | IQR1 | Ref (−) | | Ref (−) | |
| | IQR2 | 0.93 (0.80–1.09) | 0.391 | 0.91 (0.78–1.07) | 0.265 |
| | IQR3 | 1.05 (0.90–1.24) | 0.524 | 1.04 (0.88–1.22) | 0.636 |
| | IQR4 | 1.10 (0.94–1.29) | 0.228 | 1.11 (0.94–1.30) | 0.221 |
| $NO_3^-$ | IQR1 | Ref (−) | | Ref (−) | |
| | IQR2 | 0.97 (0.83–1.13) | 0.682 | 0.95 (0.81–1.11) | 0.494 |
| | IQR3 | 1.01 (0.87–1.19) | 0.868 | 1.01 (0.86–1.18) | 0.910 |
| | IQR4 | 1.11 (0.95–1.31) | 0.188 | 1.08 (0.92–1.27) | 0.365 |
| $NH_4^+$ | IQR1 | Ref (−) | | Ref (−) | |
| | IQR2 | 0.95 (0.81–1.12) | 0.537 | 0.94 (0.80–1.10) | 0.429 |
| | IQR3 | 1.02 (0.87–1.20) | 0.805 | 1.02 (0.87–1.20) | 0.809 |
| | IQR4 | 1.16 (0.99–1.36) | 0.067 | 1.11 (0.95–1.31) | 0.190 |
| OM | IQR1 | Ref (−) | | Ref (−) | |
| | IQR2 | 0.91 (0.77–1.06) | 0.222 | 0.89 (0.76–1.05) | 0.170 |
| | IQR3 | 1.01 (0.87–1.19) | 0.867 | 1.01 (0.86–1.19) | 0.876 |
| | IQR4 | 1.11 (0.94–1.30) | 0.219 | 1.10 (0.94–1.29) | 0.249 |
| BC | IQR1 | Ref (−) | | Ref (−) | |
| | IQR2 | 0.94 (0.80–1.10) | 0.431 | 0.94 (0.80–1.10) | 0.414 |
| | IQR3 | 1.06 (0.90–1.24) | 0.475 | 1.07 (0.91–1.26) | 0.419 |
| | IQR4 | 1.15 (0.98–1.35) | 0.083 | 1.16 (0.98–1.36) | 0.083 |

**Notes.**

[a] Adjusted for age, occupation, blood group, ethnicity, parity, HBV status, obesity in pregnancy, adverse reproductive history, and uterine scar.

HRs, hazard ratio; 95% CIs, 95% confidence intervals; TPROM, term premature rupture of membranes; IQR, Interquartile Range; Ref, Reference.

$PM_{2.5}$, particulate matter with aerodynamic diameter of ≤2.5 μm; $SO_4^{2-}$, sulfate; $NO_3^-$, nitrate; $NH^+$, ammonium; OM, organic matter; BC, black carbon.

to both cities being economically developed metropolitan areas with similar healthcare conditions, socioeconomic levels, and environmental factors. Additionally, although the study periods differed, their time spans were relatively close, and similarities in public health policies and living environments may also have contributed to the similarity of the results. Previous studies of the relationship between $PM_{2.5}$ and the risk of PROM have yielded inconsistent results. A retrospective study conducted in the United States revealed that the risk of PROM increased in the days or hours before delivery (*Wallace et al., 2016*). In a cohort study in Wuhan, China, involving 4,364 pregnant women, there was a positive association between $PM_{2.5}$ exposure and PROM. For each 10 μg/m$^3$ increase in $PM_{2.5}$ exposure, the risk of PROM increased by 14% (95% CI [1.02–1.26]), 9% (95% CI [1.00–1.18]), and 13% (95% CI [1.03–1.24]) in the first, second, and third

**Table 3** HRs and 95% CIs associated with TPROM in the second trimester per IQR increase in $PM_{2.5}$ and its components.

| Variable | | Crude | | Adjusted[a] | |
|---|---|---|---|---|---|
| | | HRs (95% CIs) | p-value | HRs (95% CIs) | p-value |
| $PM_{2.5}$ | IQR1 | Ref (−) | ☐ | Ref (−) | ☐ |
| | IQR2 | 1.20 (1.03–1.41) | 0.023 | 1.19 (1.01–1.40) | 0.035 |
| | IQR3 | 1.13 (0.97–1.33) | 0.121 | 1.14 (0.97–1.34) | 0.107 |
| | IQR4 | 1.02 (0.87–1.20) | 0.779 | 1.06 (0.91–1.25) | 0.459 |
| $SO_4^{2-}$ | IQR1 | Ref (−) | ☐ | Ref (−) | ☐ |
| | IQR2 | 1.17 (1.00–1.37) | 0.057 | 1.16 (0.99–1.36) | 0.069 |
| | IQR3 | 1.17 (1.00–1.37) | 0.054 | 1.17 (1.00–1.37) | 0.054 |
| | IQR4 | 1.01 (0.86–1.19) | 0.88 | 1.06 (0.90–1.24) | 0.495 |
| $NO_3^-$ | IQR1 | Ref (−) | ☐ | Ref (−) | ☐ |
| | IQR2 | 1.22 (1.04–1.43) | 0.014 | 1.19 (1.02–1.40) | 0.032 |
| | IQR3 | 1.13 (0.97–1.33) | 0.121 | 1.16 (0.99–1.36) | 0.065 |
| | IQR4 | 1.00 (0.85–1.17) | 0.967 | 1.03 (0.88–1.21) | 0.705 |
| $NH_4^+$ | IQR1 | Ref (−) | ☐ | Ref (−) | ☐ |
| | IQR2 | 1.23 (1.05–1.44) | 0.011 | 1.21 (1.03–1.45) | 0.023 |
| | IQR3 | 1.12 (0.96–1.31) | 0.160 | 1.15 (0.98–1.35) | 0.081 |
| | IQR4 | 1.01 (0.86–1.18) | 0.907 | 1.05 (0.89–1.23) | 0.577 |
| OM | IQR1 | Ref (−) | ☐ | Ref (−) | ☐ |
| | IQR2 | 1.16 (0.99–1.37) | 0.062 | 1.16 (0.99–1.36) | 0.071 |
| | IQR3 | 1.14 (0.97–1.33) | 0.115 | 1.15 (0.97–1.34) | 0.101 |
| | IQR4 | 1.06 (0.90–1.24) | 0.497 | 1.10 (0.94–1.29) | 0.248 |
| BC | IQR1 | Ref (−) | ☐ | Ref (−) | ☐ |
| | IQR2 | 1.13 (0.97–1.33) | 0.125 | 1.14 (0.97–1.33) | 0.123 |
| | IQR3 | 1.18 (1.01–1.38) | 0.044 | 1.18 (1.01–1.39) | 0.043 |
| | IQR4 | 1.05 (0.90–1.23) | 0.557 | 1.10 (0.94–1.29) | 0.247 |

**Notes.**

[a]Adjusted for age, occupation, blood group, ethnicity, parity, HBV status, obesity in pregnancy, adverse reproductive history, and uterine scar.

HRs, hazard ratio; 95% CIs, 95% confidence intervals; TPROM, term premature rupture of membranes; IQR, Interquartile Range; Ref, Reference.

$PM_{2.5}$, particulate matter with aerodynamic diameter of ≤2.5 μm; $SO_4^{2-}$, sulfate; $NO_3^-$, nitrate; $NH_4^+$, ammonium; OM, organic matter; BC, black carbon.

trimesters, respectively (*Wang et al., 2019*). However, a time series study by *Li et al. (2021)* in Shanghai, China, involving 100,200 pregnant women revealed that $PM_{2.5}$ exposure did not increase the risk of PROM. Similarly, a longitudinal study conducted in New York, USA, with 130,070 participants reported no association between $PM_{2.5}$ levels and preterm PROM (*Pereira et al., 2016*). Additionally, a study in Spain on the relationship between air pollution and preterm PROM indicated that increasing $PM_{2.5}$ exposure by one quartile did not increase the risk of preterm PROM (OR: 1.04, 95% CI [0.76–1.43]) (*Dadvand et al., 2014*). The results of this study differ from those of previous investigations. In this study, exposure to $PM_{2.5}$ during the second and third trimesters of pregnancy increased the risk of TPROM. The differences in the results may be due to several factors. First, differences in geographical regions and environmental conditions between studies could

**Table 4** HRs and 95% CIs associated with TPROM in the third trimester per IQR increase in $PM_{2.5}$ and its components.

| Variable | | Crude | | Adjusted[a] | |
|---|---|---|---|---|---|
| | | HRs (95% CIs) | *p*-value | HRs (95% CIs) | *p*-value |
| $PM_{2.5}$ | IQR1 | Ref (−) | ☐ | Ref (−) | ☐ |
| | IQR2 | 1.08 (0.93–1.27) | 0.318 | 1.08 (0.92–1.27) | 0.333 |
| | IQR3 | 1.17 (0.99–1.37) | 0.060 | 1.21 (1.03–1.42) | 0.024 |
| | IQR4 | 1.13 (0.96–1.32) | 0.145 | 1.14 (0.97–1.34) | 0.119 |
| $SO_4^{2-}$ | IQR1 | Ref (−) | ☐ | Ref (−) | ☐ |
| | IQR2 | 1.14 (0.97–1.33) | 0.114 | 1.13 (0.96–1.32) | 0.149 |
| | IQR3 | 1.14 (0.97–1.34) | 0.106 | 1.18 (1.01–1.39) | 0.040 |
| | IQR4 | 1.18 (1.00–1.38) | 0.045 | 1.18 (1.01–1.39) | 0.042 |
| $NO_3^-$ | IQR1 | Ref (−) | ☐ | Ref (−) | ☐ |
| | IQR2 | 1.07 (0.91–1.25) | 0.417 | 1.06 (0.91–1.25) | 0.452 |
| | IQR3 | 1.06 (0.91–1.24) | 0.461 | 1.07 (0.91–1.26) | 0.397 |
| | IQR4 | 1.08 (0.92–1.26) | 0.357 | 1.10 (0.94–1.29) | 0.250 |
| $NH_4^+$ | IQR1 | Ref (−) | ☐ | Ref (−) | ☐ |
| | IQR2 | 1.04 (0.89–1.22) | 0.652 | 1.03 (0.88–1.21) | 0.686 |
| | IQR3 | 1.04 (0.89–1.22) | 0.609 | 1.05 (0.90–1.24) | 0.519 |
| | IQR4 | 1.09 (0.93–1.28) | 0.285 | 1.11 (0.95–1.30) | 0.206 |
| OM | IQR1 | Ref (−) | ☐ | Ref (−) | ☐ |
| | IQR2 | 1.12 (0.96–1.31) | 0.164 | 1.12 (0.96–1.31) | 0.165 |
| | IQR3 | 1.08 (0.92–1.26) | 0.372 | 1.11 (0.94–1.30) | 0.216 |
| | IQR4 | 1.14 (0.97–1.34) | 0.105 | 1.15 (0.98–1.35) | 0.088 |
| BC | IQR1 | Ref (−) | ☐ | Ref (−) | ☐ |
| | IQR2 | 1.05 (0.90–1.23) | 0.532 | 1.04 (0.89–1.22) | 0.611 |
| | IQR3 | 1.17 (1.00–1.37) | 0.057 | 1.20 (1.03–1.41) | 0.024 |
| | IQR4 | 1.10 (0.94–1.29) | 0.224 | 1.12 (0.95–1.31) | 0.170 |

**Notes.**

[a] Adjusted for age, occupation, blood group, ethnicity, parity, HBV status, obesity in pregnancy, adverse reproductive history, and uterine scar.

HRs, hazard ratio; 95% CIs, 95% confidence intervals; TPROM, term premature rupture of membranes; IQR, Interquartile Range; Ref, Reference.

$PM_{2.5}$, particulate matter with aerodynamic diameter of ≤2.5 μm; $SO_4^{2-}$, sulfate; $NO_3^-$, nitrate; $NH_4^+$, ammonium; OM, organic matter; BC, black carbon.

lead to differences in the composition and concentration of $PM_{2.5}$, which could affect the risk of PROM. Second, differences in study methods and designs, such as sample size, exposure assessment methods and analytical techniques, could affect the accuracy of the results. For example, time series analysis was used in a study conducted in Shanghai, logistic regression models were used in Wuhan, and Cox proportional hazards models were used in this study. Finally, differences in the lifestyle, health status and genetic background of pregnant women may lead to different responses to $PM_{2.5}$ exposure.

Few studies have investigated the relationships between $PM_{2.5}$ components and PROM, particularly TPROM. *Han et al. (2020)* conducted a study in Nanjing on 1,715 pregnant women and reported that exposure to black carbon and organic matter increased the risk of PROM and shortened the gestational age. A previous study conducted in Spain on

Qin et al. (2025), *PeerJ*, DOI 10.7717/peerj.18886

**Table 5** Subgroup analysis of the associations of exposure to $PM_{2.5}$, $SO_4^{2-}$, $NO_3^-$ and TPROM during pregnancy.

| Variable | $PM_{2.5}$ | | | $SO_4^{2-}$ | | | $NO_3^-$ | | |
|---|---|---|---|---|---|---|---|---|---|
| | Adjusted HR (95% CI)[a] | *P* value | *P* for interaction | Adjusted HR (95% CI)[a] | *P* value | *P* for interaction | Adjusted HR (95% CI)[a] | *P* value | *P* for interaction |
| Age | | | 0.459 | | | 0.420 | | | 0.448 |
| >35 years | 1.01 (1.00–1.03) | 0.142 | | 1.06 (0.98–1.15) | 0.137 | | 1.04 (0.95–1.12) | 0.412 | |
| ≥35 years | 1.04 (0.98–1.11) | 0.221 | | 1.26 (0.89–1.77) | 0.187 | | 1.23 (0.84–1.80) | 0.280 | |
| Ethnicity | | | 0.821 | | | 0.713 | | | 0.626 |
| Han | 1.01 (1.00–1.03) | 0.086 | | 1.07 (0.99–1.16) | 0.084 | | 1.05 (0.97–1.14) | 0.236 | |
| Other | 1.01 (0.92–1.12) | 0.788 | | 1.13 (0.67–1.91) | 0.654 | | 0.88 (0.53–1.44) | 0.606 | |
| Occupation | | | 0.546 | | | 0.536 | | | 0.632 |
| Employed | 1.02 (1.00–1.03) | 0.107 | | 1.09 (0.99–1.20) | 0.089 | | 1.05 (0.95–1.17) | 0.317 | |
| Self-employed | 1.06 (0.98–1.15) | 0.168 | | 1.35 (0.88–2.07) | 0.166 | | 1.31 (0.82–2.10) | 0.255 | |
| Other | 1.00 (0.98–1.03) | 0.743 | | 1.02 (0.89–1.16) | 0.77 | | 1.01 (0.88–1.15) | 0.906 | |
| Blood type | | | 0.727 | | | 0.707 | | | 0.630 |
| Type A | 1.02 (0.99–1.04) | 0.260 | | 1.08 (0.94–1.24) | 0.304 | | 1.08 (0.93–1.25) | 0.322 | |
| Type B | 1.02 (0.99–1.05) | 0.210 | | 1.12 (0.95–1.32) | 0.161 | | 1.09 (0.92–1.29) | 0.340 | |
| Type O | 1.01 (0.99–1.03) | 0.375 | | 1.05 (0.94–1.18) | 0.381 | | 1.02 (0.90–1.14) | 0.779 | |
| Primary | | | 0.242 | | | 0.368 | | | 0.149 |
| No | 1.02 (1.00–1.04) | 0.030 | | 1.10 (1.01–1.2) | 0.037 | | 1.08 (0.99–1.19) | 0.096 | |
| Yes | 1.00 (0.97–1.03) | 0.873 | | 1.00 (0.85–1.18) | 0.967 | | 0.94 (0.79–1.11) | 0.447 | |
| Adverse reproductive history | | | 0.645 | | | 0.944 | | | 0.258 |
| No | 1.01 (1.00–1.03) | 0.057 | | 1.08 (1.00–1.17) | 0.065 | | 1.06 (0.98–1.15) | 0.157 | |
| Yes | 0.99 (0.94–1.05) | 0.738 | | 1.01 (0.75–1.35) | 0.964 | | 0.83 (0.61–1.14) | 0.254 | |
| Uterine scar | | | 0.883 | | | 0.964 | | | 0.842 |
| No | 1.01 (1.00–1.03) | 0.079 | | 1.08 (1.00–1.17) | 0.065 | | 1.05 (0.97–1.14) | 0.235 | |
| Yes | 1.02 (0.95–1.09) | 0.652 | | 1.05 (0.74–1.49) | 0.775 | | 0.97 (0.69–1.37) | 0.883 | |
| HBV status | | | 0.760 | | | 0.917 | | | 0.276 |
| No | 1.01 (1.00–1.03) | 0.094 | | 1.07 (0.99–1.16) | 0.079 | | 1.04 (0.95–1.12) | 0.401 | |
| Yes | 1.02 (0.96–1.10) | 0.468 | | 1.09 (0.78–1.54) | 0.608 | | 1.26 (0.87–1.84) | 0.217 | |

**Notes.**
[a] Adjusted for age, occupation, blood group, ethnicity, parity, HBV status, obesity in pregnancy, adverse reproductive history, and uterine scar.

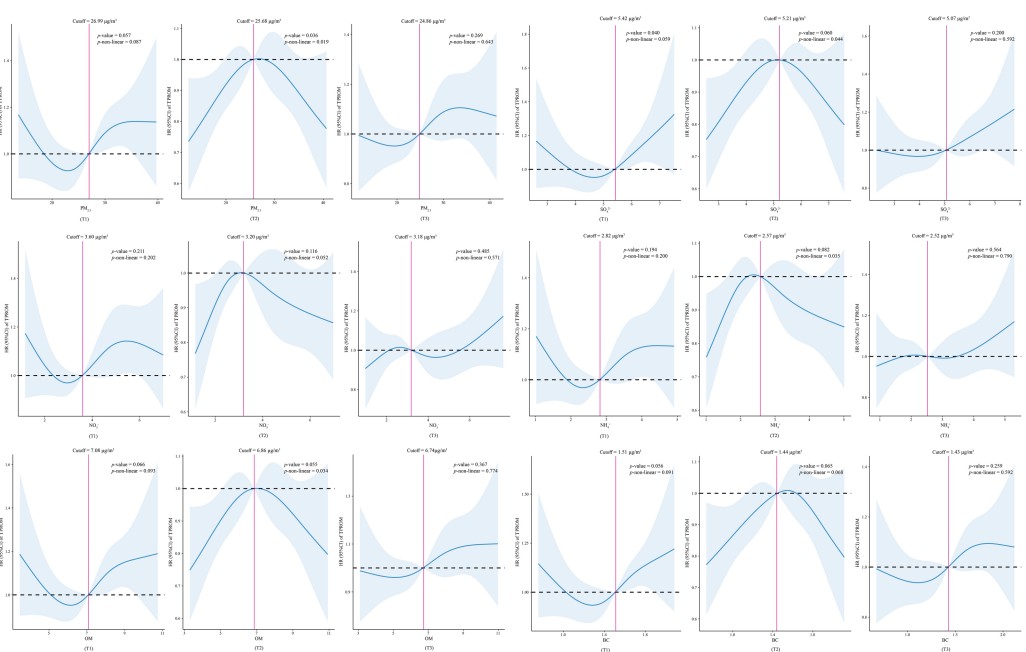

**Figure 2  Association between predicted exposure to PM$_{2.5}$ and its constituents and TPROM risk.**
T1, the first trimester. T2, the second trimester. T3, the third trimester. HR, hazard ratio. 95% CIs, 95% confidence intervals. TPROM, term premature rupture of membranes. PM$_{2.5}$, fine particulate matter. SO$_4^{2-}$, sulfate. NO$_3^-$, nitrate. NH$^+$, ammonium. OM, organic matter. BC, black carbon.

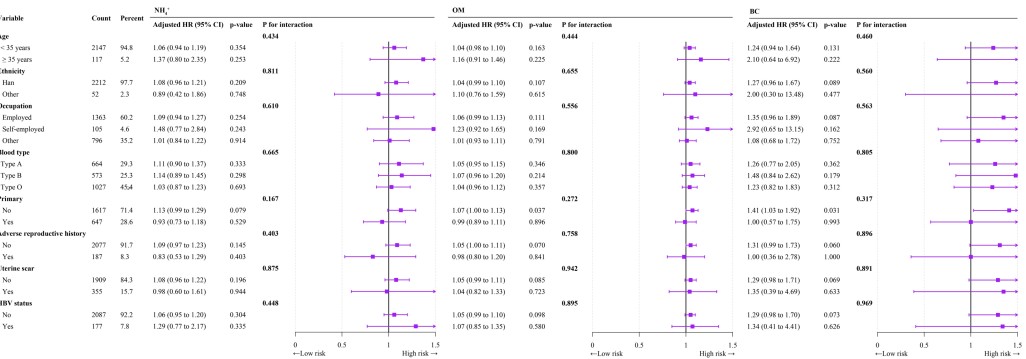

**Figure 3  Forest plot of subgroup analyses of the relationships of NH$_4^+$, OM, and BC exposure with TPROM during pregnancy.**

the relationship between air pollution and PROM indicated that for each interquartile range (IQR) increase in PM$_{2.5}$ absorbance, the risk of PROM increased by 47% (95% CI [1.08–2.00]) (*Dadvand et al., 2014*). PM$_{2.5}$ absorbance is considered a measure of black carbon (*Durant et al., 2014*). A large retrospective study conducted in the United States on prenatal exposure to air pollutants and PROM revealed that exposure to sulphates, nitrates, ammonium, and organic matter increased the risk of spontaneous PROM (*Jiao et al., 2023a*). A case-crossover study conducted in Shijiazhuang, China, on mixtures of

PM$_{2.5}$ components and TPROM revealed that exposure to SO$_4^{2-}$, NO$_3^-$, NH$_4^+$, and OM was significantly associated with an increased risk of TPROM (*Ren et al., 2024*). The results of these works differ slightly from those of this study, in which prenatal exposure to SO$_4^{2-}$, NO$_3^-$, NH$_4^+$, and BC was significantly associated with an increased risk of TPROM. Inconsistencies in results may stem from multiple factors. First, the precision of exposure assessment may be affected by the use of different assessment methods, such as monitoring station data and satellite remote-sensing data, thereby influencing the estimation of the relationship between exposure and health. Second, the biological activities of the components of PM$_{2.5}$ differ, and their direct effects on membranes and indirect effects on maternal physiology can vary depending on region and environmental conditions. Third, discrepancies in the level of control over confounding factors in studies also contribute to the variations in results. Therefore, pregnant women should therefore minimise outdoor activities during peak pollution periods and may use air purifiers to reduce indoor PM$_{2.5}$ levels. If outdoor activities in highly polluted areas are necessary, masks and other personal protective equipment should be worn to reduce the risk of inhaling harmful particulate matter. In addition, government policies can achieve long-term improvements in air quality through a range of measures, including increasing urban green spaces; strictly enforcing vehicle emission standards; restricting highly polluting industrial activities; and promoting the use of clean energy. These integrated strategies not only help improve environmental quality in general but also effectively reduce the risk of PROM in pregnant women.

This study has several strengths. First, there was a significant association between prenatal PM$_{2.5}$ exposure and preterm labour, particularly in the second and third trimesters, which improves our understanding of the effects of environmental pollution on maternal health. Second, this study considered the effects of PM$_{2.5}$ components on preterm rupture of membranes, providing insight into the roles of SO$_4^{2-}$, NO$_3^-$, NH$_4^+$ and black carbon in the risk of preterm rupture of membranes. Finally, the data support further investigations into the effects of PM$_{2.5}$ on the PROM and are consistent with findings from other cities, providing robust evidence for use by air pollution control policy-makers. However, the study has several limitations. First, the exposure assessment was based on the average exposure level at the home address, without accounting for individual exposure variations at different locations, such as daily activity spaces and workplaces. Therefore, the values may not accurately reflect the true exposure levels of individuals and could lead to biased results. Second, the analysis did not include certain health behaviours and lifestyle factors, such as smoking or cocaine use during pregnancy, as covariates, which increase the risk of PROM (*Myles et al., 1998*). Other factors, such as diet, exercise, and weight management, could also influence the results (*Faucett et al., 2016*; *Lin et al., 2024*; *Woods Jr, Plessinger & Miller, 2001*). Although age and last menstrual period were used as matching criteria to reduce confounding effects from age and gestational age, differences in gestational age might have biased the results. Finally, there are limitations regarding sample representativeness; the participants were primarily from Guangzhou, which may restrict the generalisability and applicability of the findings. The regional and socioeconomic backgrounds of the participants may not be broadly representative, affecting the applicability of the results.

Further research should incorporate participants from other regions and with other backgrounds to validate and generalise the findings.

## CONCLUSIONS

This retrospective study, which was conducted in Guangzhou, China, investigated the associations between maternal exposure to $PM_{2.5}$ and its components and TPROM. The results revealed that exposure to $PM_{2.5}$ and its components ($SO_4^{2-}$, $NO_3^-$, $NH_4^+$, and BC) significantly increased the risk of TPROM, adding new evidence to the literature and deepening our understanding of the underlying mechanisms of this adverse pregnancy outcome. Given the regional differences in air pollution levels and composition, further studies in other areas are needed to validate these findings and to explore potential regional differences.

### Funding
The authors received no funding for this work.

### Competing Interests
The authors declare there are no competing interests.

### Author Contributions
- Jiangxia Qin performed the experiments, authored or reviewed drafts of the article, and approved the final draft.
- Weiling Liu performed the experiments, analyzed the data, authored or reviewed drafts of the article, and approved the final draft.
- Haidong Zou performed the experiments, prepared figures and/or tables, and approved the final draft.
- Chong Zeng performed the experiments, prepared figures and/or tables, and approved the final draft.
- Cifeng Gao conceived and designed the experiments, performed the experiments, authored or reviewed drafts of the article, and approved the final draft.
- Weiqi Liu conceived and designed the experiments, performed the experiments, analyzed the data, prepared figures and/or tables, and approved the final draft.

### Human Ethics
The following information was supplied relating to ethical approvals (*i.e.*, approving body and any reference numbers):

The studies involving human participants were reviewed and approved by the Ethics Committee of the Maternal and Children Health Care Hospital of Huadu (approval no. 2024-001).

### Data Availability
The raw data is available in the Supplemental Files.

## Supplemental Information

Supplemental information for this article can be found online at http://dx.doi.org/10.7717/peerj.18886#supplemental-information.

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
