# Peer review of "Associations of PM2.5 and its components with term preterm rupture of membranes: a retrospective study"

_PeerJ, doi:10.7717/peerj.18886_

## Round 0.1 · original submission · Major Revisions

Dear Dr. Liu,
Your work has been evaluated by two independent experts. Both of them agreed that the paper can be published in the journal, but it needs to be revised first. Please carefully review the comments of both reviewers and address all of them.
Especially, please focus on roviding more robust statistical evidence for the matching process is important to ensure that the reduced differences between groups are not due to chance. This change will enhance the credibility of the results.
Also, please Clarifying Study Design: Addressing the confusion around the study design will ensure that the methodology is clearly understood and correctly interpreted by readers. This is fundamental for the overall validity of the study.
Knots Selection in RCS: Justifying the choice of knots will help address concerns about the model’s robustness and transparency. It is important to explain why specific percentiles were chosen, rather than using IQR, to make the methodology more reproducible.
Public Health Focus: Expanding the discussion on the public health implications of the findings will provide more context for the significance of the study, highlighting its potential impact on policy and health interventions.
One of the reviewers requested that a publication be cited. Since they are a co-author of that work, the decision to cite it (or not) is entirely up to you and will not affect the final decision regarding the publication of your paper.
With best regards,

Reviewer 1 ·

Basic reporting

The article introduced the impact of PM2.5 and its components on term premature rupture of membranes (TPROM). These authors estimated the effect using the Cox proportional hazards model and restricted cubic splines based on the case-control study. This study is important and interesting for the public health. Language is easy to understand, data analysis is accurate. However, there are currently some issues that must be addressed in the manuscript. Here are some of my specific comments.
1. Lines 148 and 151. The results mention that “After case matching, …, indicating that the differences between the case and control group were significantly reduced.” Changes in the “P” value were regarded as the standard to assess the significance of the changes in the difference between control and disease groups before and after matching in this study. Although a statistical test was not used to test this result, the disappearance of significant differences before and after matching indirectly indicated that the matching in this study was relatively successful. However, an appropriate statistical method based on hypothesis testing was used to demonstrate "significant reduced " may be more compelling evidence.
2. Lines 140-142. Restricted cubic spine is one of the common methods to explore the nonlinear relationship. The interquartile range of pollutants was used to fit the Cox proportional hazards model. However, why did the authors use the “5th, 35th, 65th, and 95th percentiles of PM2.5 and its components as knots instead of IQR (25th, 50th, 75th) in restricted cubic splines? Could the authors provide specific standards to select the number and location of knots, such as the Akaike information criterion, Bayesian information criterion, or Reference?
3. Lines 50-53. These authors described that “PROMs can be divided into preterm and term PROMs (TPROMs)” and mentioned that the objective of this study is TPROMs. Undoubtedly, TPROM is a concerning adverse pregnancy outcome. However, compared to preterm PROMs, the importance and necessity of TPROMs were not highlighted in the “Introduction”. If the authors could add some necessary references, the main objective and the logic of the “Introduction” would be clarity.
4. Lines 105-107. “Following previous research, we determined… until the birth of the baby.” The author divides pregnancy into three stages, which helps to further explore the exposure window. However, I suggest that the authors add one article (as follows) as a "previous research":
https://doi.org/10.1016/j.envres.2022.113166

Experimental design

Lines 177-179. “We investigated the associations … case-crossover analysis.” “Case-crossover” study is an epidemiological method for the impact of brief exposure on disease. This method controls the differences among individuals based on their various periods in the past, including the risk period and control period. However, the authors used a case-control study in which patients and controls were matched 1:1 for age and last menstrual period. Could the authors confirm the design type of this study, case-crossover or case-control study? Besides, “We found that exposure to … OM, and BC was significantly associated with an increased risk of PROM.” Except for nonlinear analysis, a significant association between exposure to OM and PROM was not found in this study. Adding statistical evidence or revising the description here may be considered options in this study.

Validity of the findings

no comment

Additional comments

Lines 122 and 128. These repeated the title number, please revise and check this entire text.

Reviewer 2 ·

Basic reporting

The language of the manuscript can be further polished for more concise and fluent expression.

Experimental design

no comment

Validity of the findings

no comment

Annotated reviews are not available for download in order to protect the identity of reviewers who chose to remain anonymous.

---

## Round 0.2 · accepted · Accept

Dear Dr. Liu,
One of the reviewers agreed to review your work again. Based on their opinion, and my own evaluation I was able to decide to accept the work for publication in its current version. Congratulations!

Reviewer 1 ·

Basic reporting

no comment

Experimental design

no comment

Validity of the findings

no comment

Additional comments

The authors have addressed all my concerns.